# A Knowledge Organization System for the United Nations Sustainable Development Goals

Amit Joshi[1] ✉, Luis Gonzalez Morales[1], Szymon Klarman[2], Armando Stellato[3], Aaron Helton[4], Sean Lovell[1], and Artur Haczek[2]

[1] United Nations, Department of Economic and Social Affairs, New York, NY, USA
{joshi6, gonzalezmorales, lovells}@un.org
[2] Epistemik, Warsaw, Poland {szymon.klarman, artur.haczek}@epistemik.co
[3] University of Rome Tor Vergata, Dept of Enterprise Engineering, via del Politecnico 1, 00133, Rome, Italy stellato@uniroma2.it
[4] United Nations, Dag Hammarskjöld Library, New York, NY, USA helton@un.org

**Abstract.** This paper presents a formal knowledge organization system (KOS) to represent the United Nations Sustainable Development Goals (SDGs). The SDGs are a set of objectives adopted by all United Nations member states in 2015 to achieve a better and sustainable future. The developed KOS consists of an ontology that models the core elements of the Global SDG indicator framework, which currently includes 17 Goals, 169 Targets and 231 unique indicators, as well as more than 450 related statistical data series maintained by the global statistical community to monitor progress towards the SDGs, and of a dataset containing these elements. In addition to formalizing and establishing unique identifiers for the components of the SDGs and their indicator framework, the ontology includes mappings of each goal, target, indicator and data series to relevant terms and subjects in the United Nations Bibliographic Information System (UNBIS) and the EuroVoc vocabularies, thus facilitating multilingual semantic search and content linking.

**Keywords:** United Nations, Sustainable Development Goals, SDGs, Ontology, Linked Data, Knowledge Organization Systems, Metadata

## 1 Introduction

On 25 September 2015, all members of the United Nations adopted the 2030 Agenda for Sustainable Development [11], centred around a set of 17 Sustainable Development Goals (SDGs) and 169 related Targets, which constitutes a global call for concerted action towards building an inclusive, prosperous world for present and future generations. This ambitious agenda is the world's roadmap to address, in an integrated manner, complex global challenges such as poverty, inequality, climate change, environmental degradation, and the achievement of peace and justice for all.

To follow-up and review the implementation of the 2030 Agenda, the United Nations General Assembly also adopted in 2017 a *Global SDG indicator framework* [12], developed by the Inter-Agency and Expert Group on SDG Indicators (IAEG-SDGs), as agreed earlier that year by the United Nations Statistical Commission[5], and entrusted the UN Secretariat with maintaining a Global SDG indicator database of statistical data series reported through various international agencies to track global progress towards the SDGs[6]. After its most recent comprehensive review, approved by the Statistical Commission in March 2020, the current version of the Global SDG indicator framework consists of 231 unique indicators specifically designed for monitoring each of the 169 targets of the 2030 Agenda using data produced by national statistical systems and compiled together for global reporting by various international custodian agencies. While data availability is still a challenge for some of these indicators, there are already more than 450 statistical data series linked to most of these indicators in the Global SDG Indicator database, many of which are further disaggregated by sex, age, and other important dimensions. Data from the Global SDG database is made openly available to the public through various platforms, including an Open SDG Data Hub[7] that provides web services with geo-referenced SDG indicator dataset, as well as an SDG API[8] that can support third-party applications such as dashboards and data visualizations.

In this paper, we describe the formal knowledge organization system (KOS) that has been developed for representing the United Nations Sustainable Development Goals (SDGs). In the following section, we provide the background and related linked data initiatives for the SDGs. Section 3 explains the key SDG terminologies and the ontology depicting the relationship between various SDG elements and concepts, as well as the mapping of terms and concepts in SDGs to external vocabulary and ontology. We evaluate and assess the impact of the SDG KOS in Section 4 and provide a brief explanation of LinkedSDG application in Section 5 that showcases the importance of such KOS for analysis and visualization of SDG documents. Finally, we provide the concluding remarks in Section 6.

## 2   Towards a Linked Open Data representation of the SDGs and their indicator framework

The multilingual United Nations Bibliographic Information System (UNBIS) Thesaurus [10], created by the Dag Hammarskjöld Library, contains the termi-

---

[5] This global SDG indicator framework is annually refined and subject to periodic comprehensive reviews by the UN Statistical Commission, which is the intergovernmental body where Chief Statisticians from member states oversee international statistical activities and the development and implementation of statistical standards.

[6] https://unstats.un.org/sdgs/indicators/database/

[7] https://unstats-undesa.opendata.arcgis.com/

[8] https://unstats.un.org/SDGAPI/swagger/

nology used in subject analysis of documents and other materials relevant to United Nations programme and activities. It is used as the subject authority and has been incorporated as the subject lexicon of the United Nations Official Document System. In December 2019, the Dag Hammarskjöld launched a platform for linked data services (`http://metadata.un.org`) to provide both human and machine access to the UNBIS.

At the second meeting of the Inter-Agency Expert Group on Sustainable Development Goals (IAEG-SDGs) held in Bangkok on October 2015, the UN Environment Programme (UNEP) proposed to develop an SDG Interface Ontology (SDGIO) [3], focused on the formal specification and representation of the various meanings and usages of SDG-related terms and their interrelations[9]. Subsequently, UNEP led a working group to develop the SDGIO, which either created new content or coordinated the re-use of content from existing ontologies, applying best practices in ontology development from mature work of the Open Biological and Biomedical Ontology (OBO) Foundry and Library. Currently, the SDGIO includes more than 100 terms specifying the key entities involved in the SDG process and linking them with the goals, targets, and indicators.

At its thirty-third Session held in Budapest between 30-31 March 2017, the High-level Committee on Management (HLCM) of the United Nations System's Chiefs Executives Board for Coordination adopted the UN Semantic Interoperability Framework (UNSIF) for normative and parliamentary documents, which includes Akoma Ntoso for the United Nations System (AKN4UN)[10]. Akoma Ntoso was originally developed in the context of an initiative by the UN Department of Economic and Social Affairs (UN DESA) to support the interchange and citation of documents among African parliaments and institutions and was subsequently formalized as an official OASIS standard[11].

In 2017, technical experts from across the UN System with backgrounds in library science, information architecture, semantic web, statistics and SDG indicators, agreed to initiate informal working group to develop a proposal for an "SDG Data ontology", based on the global SDG Indicator Framework adopted by the Statistical Commission[12]. The main objective of this effort was to contribute to the implementation of a linked open data approach and allow data users to more easily discover and integrate different sources of SDG-related information from across the UN System into end-user applications. The group concluded a first draft of an SDG ontology as part of the SDG Knowledge Organization System (SDG KOS), consisting of a set of permanent Uniform Resource Identifiers (URIs) for the Goals, Targets and Indicators of the 2030 Agenda and their related statistical data series based on the SKOS model.

---

[9] This includes terms such as 'access', which occurs 31 times in the SDG Global Indicator Framework.

[10] https://www.w3id.org/un/schema/akn4un/

[11] https://www.oasis-open.org/

[12] The group includes representatives from UNSD and other UN offices, including from UN DESA and the UN Library, as well as experts involved in the development of the UN Semantic Interoperability Framework adopted by the High-Level Committee on Management (HLCM) of the CEB.

In order to ensure their fullest possible use, the Identifiers and a formal Statement of Adoption were presented at the second regular session of the UN System Chief Executives Board for Coordination (CEB) in November 2019. At the CEB session, the Secretary-General invited all UN organizations to use them for mapping their SDG-related resources and sign the Statement. Subsequently, at its 51st session held in March 2020, the United Nations Statistical Commission took note of this *common Internationalized Resource Identifiers for Sustainable Development Goals, targets, indicators and related data series*, and encouraged the dissemination of data in linked open data format[13].

## 3    SDG Knowledge Organization System

### 3.1    SDG Terminologies

A sustainable development *goal* expresses an ambitious, but specific, commitment, and always starts with a verb/action. Each goal is related to a number of *targets*, which are quantifiable outcomes that contribute in major ways to the achievement of the corresponding goal. An *indicator*, in turn, is a precise metric to assess whether a target is being met. There may be more than one indicator associated with each target. In rare cases, the same *indicator* can belong to multiple *targets*. The global indicator framework lists 247 indicators but has only 231 unique indicators[14].

Finally, a *series* is a set of observations on a quantitative characteristic that provides concrete measurements for an *indicator*. Each *series* contains multiple records of data points organized over time, geographic areas, and/or other dimensions of interest (such as sex, age group, etc.).

To facilitate the implementation of the global indicator framework, all indicators are classified into three *tiers* based on their level of methodological development and the availability of data at the global level, as follows:

- Tier I: Indicator is conceptually clear, has an internationally established methodology and standards are available, and data are regularly produced by countries.
- Tier II: Indicator is conceptually clear, but data are not regularly produced by countries.
- Tier III: No internationally established methodology or standards are yet available for the indicator.

The updated tier classification contains 130 Tier I indicators, 97 Tier II indicators and 4 indicators that have multiple tiers (different components of the indicator are classified into different tiers)[15].

---

[13] See https://www.undocs.org/en/E/CN.3/2020/37 - United Nations Statistical Commission, Decision 51/102 (g) on Data and indicators for the 2030 Agenda for Sustainable Development.

[14] See https://unstats.un.org/sdgs/indicators/indicators-list for the updated list of indicators that repeat under two or three targets

[15] List is regularly updated and available at https://www.undocs.org/en/E/CN.3/2020/37.

### 3.2   Namespaces

The namespace for SDG ontology is `http://metadata.un.org/sdg/`. The ontology extends and reuses terms from several other vocabularies in addition to UNBIS and EuroVoc. A full set of namespaces and prefixes used in this ontology is shown in Table 1.

Table 1: Namespaces used in SDG Ontology

| Prefix | Namespace |
|--------|-----------|
| sdgo: | http://metadata.un.org/sdg/ontology |
| dc: | http://purl.org/dc/elements/1.1/ |
| owl: | http://www.w3.org/2002/07/owl# |
| rdf: | http://www.w3.org/1999/02/22-rdf-syntax-ns# |
| rdfs: | http://www.w3.org/2000/01/rdf-schema# |
| xsd: | http://www.w3.org/2001/XMLSchema# |
| skos: | http://www.w3.org/2004/02/skos/core# |
| wd: | http://www.wikidata.org/entity/ |
| sdgio: | http://purl.unep.org/sdg/ |
| unbis: | http://metadata.un.org/thesaurus/ |
| ev: | http://eurovoc.europa.eu/ |
| dct: | http://purl.org/dc/terms/ |

### 3.3   Schema

The SDG ontology formalizes the core schema of SDG goal-target-indicator-series hierarchy, consisting of four main classes namely *sdgo:Goal*, *sdgo:Target*, *sdgo:Indicator* and *sdgo:Series*, which correspond to the four levels of the SDG hierarchy, and three matching pairs of inverse properties – one per each level, as shown in Figure 1.

The properties are further constrained by standard domain and range restrictions. While logically shallow, such axiomatization avoids unnecessary overcommitment and guarantees good interoperability across different ontology management tools.

Furthermore, the SDG ontology is aligned with (by extending it) the SKOS Core Vocabulary[16], a lightweight W3C standard RDF for representing taxonomies, thesauri and other types of controlled vocabulary [8]. This alignment rests upon the set of sub-class and sub-property axioms as shown in Figure 1, which additionally emphasizes the strictly hierarchical structure of the SDG entities. The top concepts in the resulting SKOS concept scheme are the SDG goals, while the narrower concepts, organized into three subsequent levels, include targets, indicators, and series, respectively.

The choice of SKOS is not restrictive on the nature of SDG items: SKOS and specific OWL constructs can be intertwined into elaborated ontologies that guarantee both strict adherence to a domain (through the definition of domain-oriented properties, such as the aforementioned properties linking the strict 4-layered architecture of the SDGs) and interoperability on a coarser level. For

---

[16] https://www.w3.org/2009/08/skos-reference/skos.html

instance, the use of *skos:narrower* and *skos:broader* allows any SKOS-compliant consumer to properly interpret the various levels of the SDG as a hierarchy and to show it accordingly.

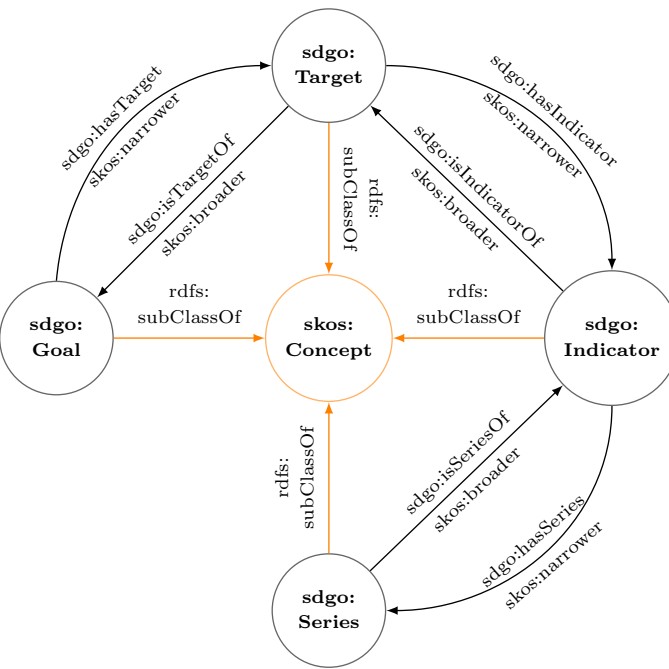

Fig. 1: Core structure of the SDG ontology modelled using SKOS vocabulary

Following from the SKOS specifications, the *skos:broader/narrower* relation is not meant to imply any sort of "subclassification" (e.g. according to the set-oriented semantics of OWL) and indeed it should not for two reasons: a) extensionally, each object in the SDG, be it a goal, target or indicator, is not a class of objects, but rather a specific object itself; b) the SKOS hierarchy does not suggest any sort of specialization, as the relations among the various levels are closer to different nuances of a dependency relation. SKOS properties *skos:broader/narrower* (their name could be misleading) represent a relation meant exclusively for depicting hierarchies, with no assumption on their semantics (see, for instance, the possibility to address a part-of relation as indicated in the SKOS-primer[17]). This is perfectly fitting for the SDG representation, as SDGs are always disseminated as a taxonomy.

Finally, the SDG ontology also covers a three-level Tier classification for Global SDG Indicators, which supports additional qualification of the indicators in terms of their implementation maturity.

---

[17] https://www.w3.org/TR/skos-primer/#sechierarchy

### 3.4   SDG Data

The SDG data, analogously to the schema-level, is represented in terms of two terminologies: the native SDG ontology and the SKOS vocabulary. Every instance is described as a corresponding SDG element and a SKOS concept. As an example, Table 2 presents data pertaining to the SDG Target 1 of Goal 1. The labels and values of *skos:notation* property reflect the official naming and classification codes of each element.

Table 2: RDF Description of SDG Target 1.1

| subject = http://metadata.un.org/sdg/1.1 | |
| --- | --- |
| **predicate** | **object** |
| rdf:type | skos:Concept |
| rdf:type | sdgo:Target |
| skos:inScheme | http://metadata.un.org/sdg |
| skos:broader | :1 |
| sdgo:isTargetOf | :1 |
| skos:narrower | :C010101 |
| sdgo:hasIndicator | :C010101 |
| skos:prefLabel | "By 2030, eradicate extreme poverty for all people everywhere, currently measured as people living on less than $1.25 a day"@en |
| skos:prefLabel | "D'ici à 2030, éliminer complètement l'extrême pauvreté dans le monde entier (s'entend actuellement du fait de vivre avec moins de 1,25 dollar des États-Unis par jour)"@fr |
| skos:prefLabel | "De aquí a 2030, erradicar para todas las personas y en todo el mundo la pobreza extrema (actualmente se considera que sufren pobreza extrema las personas que viven con menos de 1,25 dólares de los Estados Unidos al día)"@es |
| skos:notation | "1.1"^^sdgo:SDGCodeCompact |
| skos:notation | "01.01"^^sdgo:SDGCode |
| skos:note | "Target 1.1"@en |
| skos:note | "Cible 1.1"@fr |
| skos:note | "Meta 1.1"@es |

The codes are represented in two notational variants catering for different presentation and data reconciliation requirements. In several cases the indicators and series have more than one broader concept in the hierarchy, as defined in the Global indicator framework. For instance, the indicator sdg:C200303 appears as narrower concept (*sdgo:isIndicatorOf*) of targets 1.5, 11.5 and 13.1, and is consequently equipped with three different sets of codes: "01.05.01", "1.5.1", "11.05.01", "11.5.1", "13.01.01", "13.1.1". Due to this poly-hierarchy, the indicators are equipped with an additional set of unique identifier codes of the form "*Cxxxxxx*". Table 3 shows the partial RDF description of this indicator C200303.

The unique identifiers of the SDG series, reflected in their URIs and in the values of the *skos:notation* property, follow the official coding system of the UN Statistics Division[18]. As the list of relevant SDG series is subject to recurrent updates, their most recent version captured in the published SDG ontology is

---

[18] https://unstats.un.org/sdgs/indicators/database/

Table 3: RDF Description of SDG Indicator C200303 depicting poly-hierarchy

| subject = http://metadata.un.org/sdg/C200303 | |
|---|---|
| **predicate** | **object** |
| rdf:type | skos:Concept
sdgo:Indicator |
| skos:inScheme | http://metadata.un.org/sdg |
| skos:broader | :1.5
:11.5
:13.1 |
| sdgo:isIndicatorOf | :1.5
:11.5
:13.1 |
| skos:narrower | sdg:VC_DSR_AFFCT
sdg:VC_DSR_DAFF
sdg:VC_DSR_DDHN |
| sdgo:hasSeries | sdg:VC_DSR_AFFCT
sdg:VC_DSR_DAFF
sdg:VC_DSR_DDHN |
| sdgo:tier | sdgo#tier_II |
| skos:prefLabel | "Number of deaths, missing persons and directly affected persons attributed to disasters per 100,000 population"@en |
| skos:notation | "1.5.1"^^sdgo:SDGCodeCompact
"01.05.01"^^sdgo:SDGCode
"11.5.1"^^sdgo:SDGCodeCompact
"11.05.01"^^sdgo:SDGCode
"13.1.1"^^sdgo:SDGCodeCompact
"13.01.01"^^sdgo:SDGCode
"C200303"^^sdgo:SDGPerm |
| skos:exactMatch | http://purl.unep.org/sdg/SDGIO_00020006 |
| skos:custodianAgency | "UNDRR" |

represented under the *skos:historyNote* property ("2019.Q2.G.01" at the time of writing).

### 3.5   Linking the SDG Ontology

The SDG ontology has been enriched with additional mappings to existing vocabularies and ontologies in order to facilitate content cataloguing, semantic search, and content linking. Specifically, each goal, target and indicator has been mapped with topics and concepts defined in UNBIS and EuroVoc[19] thesauri. Identifiers have also been mapped to external ontologies like SDGIO and Wikidata[20]. Wikidata is a free and open structured knowledge base that can be read and edited by both humans and machines [4]. Figure 2 depicts association of sdg:1 to wikidata and SDGIO via *skos:exactMatch*, as well as concept mappings to UNBIS and EuroVoc via *dct:subject*.

Mapping the SDG elements to both UNBIS and EuroVoc thesauri enables knowledge discovery as both contain large number of concepts across many domains. The UNBIS Thesaurus contains more than 7000 concepts across 18 domains and 143 micro-thesauri with lexicalizations in the six official languages

---

[19] https://op.europa.eu/en/web/eu-vocabularies
[20] https://www.wikidata.org/

of the UN, providing one of the most complete multilingual resources in the organization. EuroVoc is European Union's multilingual and multidisciplinary thesaurus that contains more than 7000 terms, organized in 21 domains and 127 sub-domains. Section 5 provides further details on the importance of linking to external vocabulary for extracting relevant SDGs.

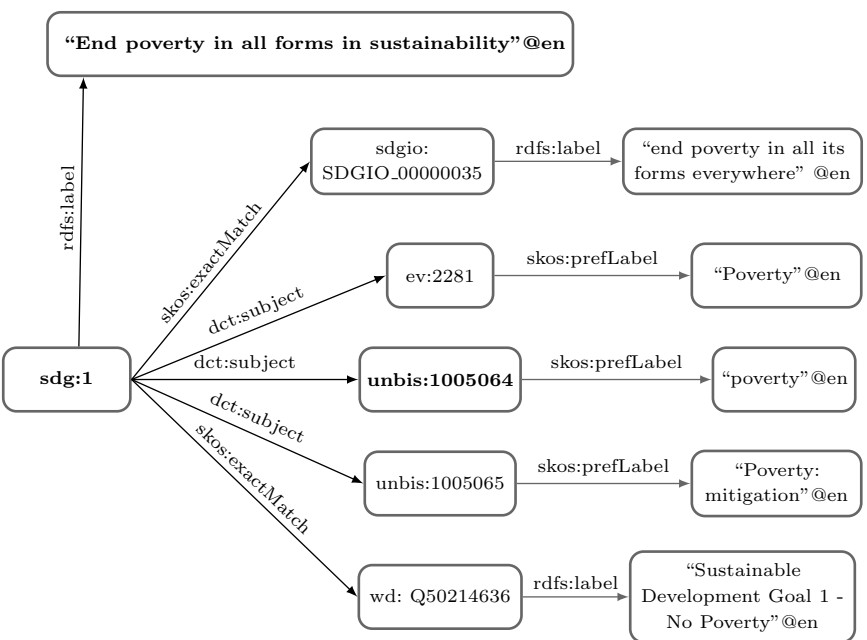

Fig. 2: SDG Goal#1 linked to other vocabularies including UNBIS and EuroVoc.

## 4   Evaluation and Impact of the SDG

In this section we evaluate and assess the impact of the SDG KOS under several dimensions, including usability, logical consistency and support of multilingualism.

### 4.1   Usability

The resource is mainly modelled as a SKOS concept scheme, for ease of consumption and interpretation of the SDG taxonomy by SKOS-compliant tools. As a complement to that, the ontology vocabulary provides different subclasses of the *skos:Concept* (adopted in the KOS) for clearly distinguishing the nature of Goals, Targets, Indicators and Series, and subproperties of *skos:broader* and

*skos:narrower*. OWL axioms constrain the hierarchical relationships among them (for example, Targets can only be under Goals and have Indicators as narrower concepts).

Both the ontology and the SKOS content are served through multilingual descriptors (see section 4.4), thus facilitating interpretation of the content across various idioms and annotation of textual content with respect to these global indicators. *skos:notations* are provided in three different formats (according to various ways in which the global indicators codes have been rendered). For each format, a different datatype has been coined and adopted, so that platforms with advanced rendering mechanisms (e.g. VocBench [9]) can select the proper one and use it as a prefix in the rendering of each global indicator (e.g. ⟨notation⟩ and ⟨description⟩).

### 4.2   Resource Availability

The SDGs are available on the site of the statistical division of the UN through a web interface[21], allowing for the exploration of their content, and as Excel and PDF files. The Linked Open Data (LOD) version of the dataset is available on the metadata portal of the UN, with separate entries for the ontology[22] and the taxonomy of global indicators[23]. The namespace of SDG ontology and KOS have been introduced in section 3. As a large organization with a solid management of its domain, the LOD version of the dataset does not rely on an external persistent URI service (e.g. PURL, DOI, W3ID) as the metadata.un.org URIs can be considered persistent and reliable. All URIs resolve through content negotiation. The ontology offers a web page for human consumption describing its content, resolving to the ontology file[24] in case of requests for RDF data. The KOS provides an explorable taxonomy on a web interface at the URI of the concept scheme and human-readable descriptions (generated after the RDF data) of each single global indicator at their URIs. All of them http-resolve to RDF in several serialization formats (currently Turtle, RDF/XML, NT and JSON-LD).

The project related to the development of the SDG data and the data itself (available as download dumps) are available under a public GitHub repository[25].

Sustainability. SDGs are a long-term vision of the United Nations with an end date of year 2030. As such, the SDG KOS and associated tools and libraries will be actively maintained and updated over the years and are expected to be reachable even in case the SDG mission is considered concluded.

Licensing: The SDG Knowledge Organization System is made available free of charge and may be copied freely, duplicated and further distributed provided that a proper citation is provided. License information is also available on its VoID descriptors.

---

[21] https://unstats.un.org/sdgs/indicators/database/
[22] http://metadata.un.org/sdg/ontology
[23] http://metadata.un.org/sdg/
[24] e.g. http://metadata.un.org/sdg/static/sdgs-ontology.ttl for Turtle format
[25] https://github.com/UNStats/LOD4Stats/wiki

### 4.3 VoID/LIME Description

Metadata about the linked open dataset has been reported in a machine-accessible VoID [1] and LIME [5] files at UN Metadata site[26]. The VoID file also contains entries linking the download dumps.

VoID is an RDF vocabulary for describing linked datasets, which has become a W3C Interest Group Note[27]. VoID provides the policies for its publication and linking to the data [6] and also defines a protocol to publish dataset metadata alongside the actual data, making it possible for consumers to discover the dataset description just after encountering a resource in a dataset. Developed within the scope of the OntoLex W3C Community Group[28], LIME is an extension of VoID for linguistic metadata. While being initially developed as the metadata module of the OntoLex-Lemon model[29] [7], LIME intentionally provides descriptors that can be adapted to different scenarios (e.g. ontologies or thesauri being lexicalized, resources being onomasiologically or semasiologically conceived) and models adopted for the lexicalization work (*rdfs:labels*, SKOS or SKOS-XL terminological labels or Ontolex lexical entries).

The VoID description is organized by first providing general information about the dataset through the usual Dublin core [2] properties, such as description, creator, date of publication, etc. Of particular notice is the dct:conformsTo property, pointing in this case to the SKOS namespace and which can be adopted by metadata consumers in order to understand the core modelling vocabularies being adopted representing the dataset. The description is followed by a few *void:classPartitions* providing statistics about the types of resource characterizing the type of dataset (as informed by the aforementioned *dct:conformsTo*).

In the case of SDG, the template for SKOS has therefore been applied, providing statistics for *skos:Concepts*, *skos:Collections* and *skos:ConceptSchemes*, reporting a total of 812 concepts, 1 concept scheme and 1 collection. The description continues with typical *void* information, such as statistics about the number of distinct subjects (1002), objects (6842), triples (14645), availability of a SPARQL endpoint and downloadable data dump (*void:dataDump*), which we provided for the full dump as well as for some partitioned versions of the dataset (e.g. ontology only, ⟨ Goal, Target, Indicator ⟩ only, etc..). Finally, a list of subsets are then described in detail in the rest of the file. This list is mainly composed of *void:Linksets*, which are datasets consisting of a series of alignment triples between the described dataset and other target datasets, and of *lime:Lexicalizations*, the portions of the described dataset containing all the triples related to the (possibly multiple, as in this case) lexicalizations that are available for it. Each lexicalization is described in terms of its lexicalization model (which in the case of the SDGs is SKOS), of the natural language covered by the lexicalization, expressed in terms of ISO639-1 2-digit code as a literal (through property *lime:language*), and of ISO639-1 2-digit code and ISO639-3

---

[26] http://metadata.un.org/sdg/void.ttl
[27] http://www.w3.org/TR/void/
[28] https://www.w3.org/community/ontolex/
[29] https://www.w3.org/2016/05/ontolex/

3-digit code in the form of URIs using the vocabulary of languages[30] of the Library of Congress[31]. It also includes information about the lexicalized dataset (*lime:referenceDataset*) and void-like statistical information such as the total number of lexicalized references, the number of lexicalizations, the average number of lexicalizations per reference and the percentage of lexicalized references. More details about the available lexicalizations will be provided in section 4.4 on multilingualism.

### 4.4    Multilingualism

As the SDGs are a United Nations resource, multilingualism is an important factor and a goal to be achieved. Currently, the resource is available in three languages: English, French and Spanish. As reported in the LIME metadata within the VoID file, English is currently the most represented language (being the one natively used for redacting the SDGs), with 99.9% of resources (813 out of 814) covered by at least a lexicalization (which means it includes the Series linked to the SDGs as well) and an average number of 1,021 lexicalizations per resource, thus allowing for some cases of synonymy. French and Spanish follow almost equally with 422 and 420 lexicalizations respectively covered (roughly 51.4% of the resources). This is approximately equal to the total number of Goals, Targets and Indicators combined, and excludes the Series which still have to be translated to these languages. Other languages are being planned to be added to the list of lexicalizations.

### 4.5    A Comparison with the SDGIO Ontology

The SDGIO claims to be an Interface Ontology (IO) for SDGs, providing "a semantic bridge between 1) the Sustainable Development Goals, their targets, and indicators and 2) the large array of entities they refer to" . To achieve this objective, it "imports classes from numerous existing ontologies and maps to vocabularies such as GEMET to promote interoperability". Among various connections, SDGIO strongly builds on multiple OBO Foundry ontologies "to help link data products to the SDGs". The SDGIO offers a very specific interpretation, where the various Goals, Targets and Indicators are instances of their respective classes and then Indicators have also a "sustainable development goal indicator value" class (which contains, as subclasses, the various indicators) to contain values for these indicators. The SDGIO does not include any information about the Series. Differently from the objectives of SDGIO, which is focused on its interfacing aspect and is based on a strong commitment to OBO ontologies, our mission was to build the official representation of SDGs and to place it under a largely interoperable, yet neutral, perspective. To this end, we represented SDG elements as ground objects, providing an ontology for describing their nature and mutual relationships, which builds in turn on top of the SKOS vocabulary, for purposes of visualization in SKOS-compliant consumers.

---

[30] http://id.loc.gov/vocabulary/
[31] https://loc.gov/

## 5   Knowledge Discovery and Linking SDGs

The SDG ontology is part of an emerging system of SDG-related ontologies that aim to provide data inter-operability and a flexible interface for querying linkages across independent information systems. Mapping the identifiers described in these ontologies to each other and to external vocabularies allows SDG data to be clearly identified and found by semantic web agents for establishing further links and connections thereby facilitating knowledge discovery.

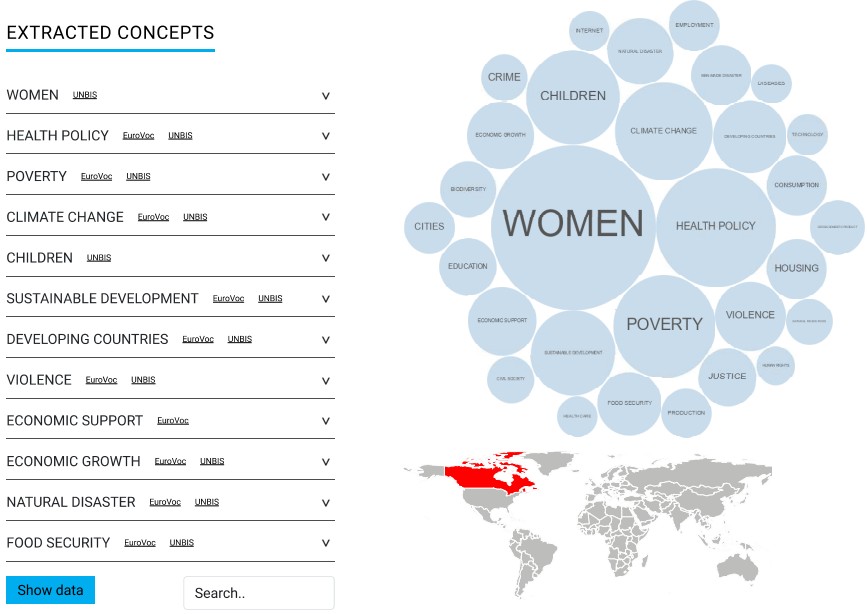

Fig. 3: Application displaying extracted concepts and corresponding tag cloud, and a map highlighted with geographic regions mentioned in a document.

A pilot application, LinkedSDG[32], has been built to showcase the usefulness of adopting SDG KOS for extracting SDG related metadata from documents and establishing the connections among various SDGs. The application automatically extracts relevant SDG concepts mentioned in a given document using SDG KOS and provides their unified overview. All SDGs related to the identified concepts are displayed in an interactive wheel chart that users can further explore by drilling into associated goals, targets, indicators and series. Figure 3 and Figure 4 depicts different components of application including concept extraction, map,

---

[32] http://linkedsdg.apps.officialstatistics.org/, source code for the application available at: https://github.com/UNGlobalPlatform/linkedsdg.

SDG wheel chart and associated data series for one of the Voluntary National Reviews (VNRs)[33]. The application can process documents written in any one of the six UN official languages.

**MOST RELEVANT SDGS**

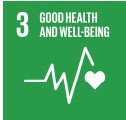 Ensure healthy lives and promote well-being for all at all ages

**NAME**: Target 3.3

**DESCRIPTION**: By 2030, end the epidemics of AIDS, tuberculosis, malaria and neglected tropical diseases and combat hepatitis, water-borne diseases and other communicable diseases

URI: http://metadata.un.org/sdg/3.3

**KEYWORDS:**

**HEALTH**   UNBIS

      MENTAL HEALTH        EuroVoc UNBIS
      PUBLIC HEALTH         EuroVoc UNBIS
      HEALTH CONTROL       EuroVoc UNBIS

**COMMUNICABLE DISEASES** UNBIS

      TUBERCULOSIS         UNBIS
      AIDS                 EuroVoc UNBIS
      POLIOMYELITIS        UNBIS

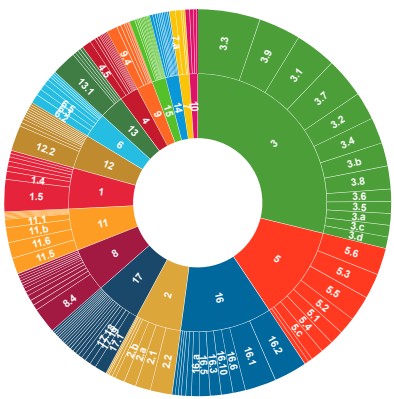

**DATA SERIES**

Tuberculosis incidence (per 100,000 population)

| country | year | value |
|---------|------|-------|
| Canada  | 2000 | 6.4   |
| Canada  | 2001 | 6.6   |
| Canada  | 2002 | 6.1   |

1 2 3 4 5 > >>

Fig. 4: Application showing the most relevant SDG keywords corresponding to Target 3.3, an interactive SDG wheel corresponding to a document and related data series for one of the Indicators

The two key analytical techniques employed in LinkedSDG are taxonomy-based *term extraction* and *knowledge graph traversal*. The term extraction mechanism, implemented using the *spaCy* library [34], scans the submitted document for all literal mentions of the relevant UNBIS and EuroVoc concept labels, based on the initially detected language of the document, and associates them with their respective concept identifiers. The traversal, performed using SPARQL via the underlying *Apache Jena* RDF store[35], starts from these extracted concept identifiers, following to broader ones, to finally reach those connected directly to the elements of the SDG system via the *dct:subject* and *skos:exactMatch* predicates. Then, the algorithm traces the paths to broader SDG entities in the SDG KOS hierarchy. For instance:

---

[33] See https://sustainabledevelopment.un.org/ for VNR documents

[34] https://spacy.io/

[35] https://jena.apache.org/

- **text**: "[...] beaches, estuaries, dune systems, mangroves, marshes lagoons, swamps, reefs, etc are [...]"
- **extracted concept**: WETLANDS (unbis:1007000) via the matched synonym "marshes"
- **traversed path**: WETLANDS - (broader) → SURFACE WATERS (unbis:1006307) - (broader) → WATER (unbis:030500)
- **connected target**: 6.5 By 2030, implement integrated water resources management at all levels, including through transboundary cooperation as appropriate
- **connected goal**: 6. Clean water and sanitation

The computation of the final relevance scores for specific goals, targets and indicators relies on their exact positioning in the SDG hierarchy, which is reflected by the SKOS representation of the system, and on their types, asserted in the SDG ontology. Intuitively, the broader the terms (i.e., the higher in the hierarchy) the higher score they receive, as they aggregate the scores in the lower parts of the hierarchy.

The application also provides access to the statistical data of the specific SDG series, which is represented as linked open statistical data using the *RDF Data Cube* vocabulary[36]. The relevant SDG series identifiers are referenced from the extraction results delivered by the application and independently served by the platform's dedicated GraphQL API [37]. Consequently, SDG KOS fueling the LinkedSDG platform supports the user in the entire journey from a text document to the relevant statistics, helping put the originally unstructured, third-party information, in the context of narrowly focused, UN-owned structured data.

## 6    Conclusion

The integration of multiple sources and types of data and information is fundamental to guide policies aimed to achieve the 2030 Agenda for Sustainable Development. The complexity and scale of the global challenges require solutions that take into account trade-offs and synergies across the social and economic dimensions of sustainable development. In order to foster a holistic approach through coordinated policies and actions that bring together different levels of government and actors from all sectors of society, it is crucial to develop tools that facilitate the discovery and analysis of interlinkage across various global SDG indicators, as well as across other sources of data, information and knowledge maintained by different stakeholder groups.

The SDG KOS is an attempt to provide stakeholders a means to publish the data using common terminologies and URIs centred around the SDG concepts, thus helping break information silos, promote synergies among communities, and enhance the semantic interoperability of different SDG-related data and information assets made available by various sectors of society.

---

[36] https://www.w3.org/TR/vocab-data-cube/
[37] http://linkedsdg.apps.officialstatistics.org/graphql/

**Acknowledgements**

Much of the work towards developing the SDG KOS and the LinkedSDG pilot application was conducted in the context of a UN DESA project funded through the EU grant entitled "SD2015: Delivering on the promise of the SDGs". The authors would like to acknowledge the invaluable contributions and guidance from Naiara Garcia Da Costa Chaves, Susan Hussein, Flavio Zeni, as well as from many other colleagues from UN DESA, the Dag Hammarskjöld Library, and the Secretariat of the High Level Committee on Management of the UN Chief Executive Board for Coordination.

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
