# OpenReview forum: "A Knowledge Organization System for the United Nations Sustainable Development Goals"
_eswc-conferences.org/ESWC/2021/Conference/Resources_Track — ESWC 2021 Resources_

### Official Review · AnonReviewer3 · 2021-01-14
**A Knowledge Organization System for the United Nations Sustainable Development Goals**

**Rating:** 1
**Confidence:** 4

**Review:**

This paper presents an ontology about sustainable development for the United Nations Organization.
The ontology is part of an information system with databases about statistical data about economical and social development all over the world.

The ontology topic is the definition of Goal, Target, Indicator and Series concepts in relation with other models and databases which form a digital ecosystem.

The structure of the ontology is specific as it relies both on SKOS and RDFS using puning.

The four main concepts are subClassOf skos:Concept and their instances are organized using skos:broader/narrower, hence  the ontology is a SKOS thesaurus.

On another hand there are three specific properties relating the four kinds of concepts: sdgo:hasTarget, sdgo:hasIndicator, sdgo:hasSeries and these properties are subPropertyOf of skos:narrower.

Hence the ontology describes a data model and it is also a SKOS thesaurus.

This specific point is not explicitely clearly written in the paper.

The paper  describes at length the structure of the digital ecosystem as well as the commitees that manage the ecosystem. This is not the most interesting part for the reader.
On another hand the content of the ontology is not detailed which is a bit frustrating.


**Anonymity:**

Yes, I would like my review to remain anonymous.

**Strong Points:**

Interesting resource on a general interest important topic
Ontology with a specific architecture.

**Subreviewer:**

I submitted this review.

**Weak Points:**

The paper does not focus on the content of the ontology but on management and organizational issues

---

> ### Author Rebuttal · Authors · 2021-01-29
>
> We thank the reviewer for the feedback. We acknowledge their demand for more details about the content: we indeed focused more on the representation of the SDGs for semantic standards and on the general description of the datasets, moving to then to the background that led to their development as an RDF dataset. Perhaps proper references to literature detailing the SDGs content would be appropriate, as it would take too much space talking more about the content, subtracting it from the one reserved to it as a linked open dataset.
>
> One remark: there is no punning. Goal, Target etc.. are not concepts (as in the sense of skos:Concept)  rather classes, subclasses of skos:Concept. Their instances are thus individuals (as instances of class skos:Concept always are) so everything stays well within the boundaries of a (subset of) first-order logics. About the data model: SKOS is a rich model, which has not to be limited to mere broader/narrower relations and terminological properties for labeling. “Subproperting” SKOS properties to those of a domain (so to better evoke the relations in that domain) is a well-known pattern, in this case meant to put a clear separation between the levels in the hierarchy; e.g. in plain SKOS each concept in the various levels is just a concept, while in this case the first layer is Goals, then Targets, Indicators, Series and the domain and range of the domain-specific properties enforce this strict layering. The same could have been obtained with class axioms on the Goal, Target, etc.. classes restricting the use of the sole broader/narrower properties. Indeed, this has been a first modeling of the SDGO ontology and it has been a subject of discussion: the final choice was only meant to provide a more evident distinction yet from the name of the properties, whereas in the former case this would have been enforced only in logical terms. Whichever the way, the system is still a thesaurus, as those classes describe no other data than the concepts. However, if the reviewer think this appropriate, we can introduce this rationale (it is already mentioned in section 3.3, start of page 6, but we can further elaborate on it).

---

> > ### Comment · AnonReviewer3 · 2021-02-01
> > **Rebuttal**
> >
> > The reviewer acknowledge the rebuttal.
> > The reviewer think you must introduce the rationale for mixing SKOS and RDFS.

---

### Official Review · AnonReviewer1 · 2021-01-16
**Good work**

**Rating:** 2
**Confidence:** 5

**Review:**

The paper describes the developments of the ontology, data and applications about UN Sustainable Development Goals.

The work is carefully and professionally done, with the open schemas, data and applications available for everyone's use. The paper is well written, and the descriptions are clear.

The developments are novel and apply well the state of the art in the field. They are also significant, due to the importance of the domain, and broad interest to the topic in the years to come.

A few things that should be looked at to improve the paper are:
- In section 3.4, the table 3 should appear after the reference to "Table 3",
- in section 3.5, it would be good to explain the approach that was used to create the mappings. They were not made randomly, or? Or is the set of the mappings in some way complete?
- the end of section 4.2 is strangely presented with single words (not sentences) such as "Sustainability.", "Licensing." inside the text. Generally these points do not have a large connection to the main topic of the section, which is availability. It would be of added value to explain and approach the evaluation more systematically, possibly following some existing methodologies for it (e.g. on data quality, FAIR data assessment). This may result in changes in the sections structure here.

Typos:
- page 1: the "1" at the end of the 2nd author's surname,
- page 3: in the references at the footnotes, some sentences are finished with the full stop and some not,
- page 6: check the grammar and punctuation in the sentence starting with "For instance: The use of skos:...",
- page 9: "... them so that, say, the ontology..." - "say" is out of style here, can be changed, for example to "for example"; similar with "info" on page 10,
- page 10: check the punctuation on the last line,
- page 11: the sentence "Finally, a list of subsets..." does not have a verb. Also, check the punctuation of the sentence before it.

**Anonymity:**

No, I would like my review to be deanonymized.

**Strong Points:**

- Official open government data on relevant domain.
- Uniqueness.
- Quality of work.

**Subreviewer:**

I submitted this review.

**Weak Points:**

- Current application demonstrate visualisation aspects only, it would be useful to describe what the approach to engage other developers and users with the developed system is.

---

> ### Author Rebuttal · Authors · 2021-01-29
>
> We thank the reviewer for the feedback. We acknowledge - as this has been also a common observation shared by other reviewers - and fully agree with the request on more insights about mappings and connection with existing methodologies (FAIR, thus including even compliancy with its requirements).
> We also acknowledge all  the typos (thanks once more) and will apply the suggested fixes.

---

> > ### Comment · AnonReviewer1 · 2021-01-30
> > **Rebuttal acknowledgement**
> >
> > I acknowledge reading the rebuttal. Good luck with further work.

---

### Official Review · AnonReviewer4 · 2021-01-17
**Ontology for SDG supported by UN**

**Confidence:** 5

**Review:**

I've read the response and disagree with many points about the use of SKOS. Some claims (for example about the punning) also apply in the other direction. The same with the mappings, if their use don't harm why using the skos with no skos concepts and other mappings with SKOS concepts. I would undersating using the same mappings to all but exchanging models has no sense.

#

The resource presented fills an important gap and it is likely to be broadly reused as reference for SDOs definitions. In general meets most of the requirements set for the resources track. One requirement not fulfilled is the used of Purl or w3id however I think it is well justified as United Nations URIs are not likely to disappear and it reinforce the authority of the resource, so a well-defined strategy within UN would be enough to guarantee persistence. Actually it is very encouraging seeing organizations as UN adopting semantic web technologies.

There are however some shortcoming from the design and technical aspects that I would like to point and see authors actions/responses before fully accepting the paper.

Main issue with the presented ontology and dataset is the decision about using SKOS concepts for the modelling rather than owl classes. I could understand the use of SKOS for Goal, Target and Indicator as kind of thesaurus, even though it would be more natural for me to model them just as classes and domain object properties. Apart from that, it is more difficult to understand the decision for "Series", in this case I see more clear that the relation between indicator and series is not from a KOS system but a domain relation.

Another important issue is about the mappings with external resources. Why using exactMatch with wikidata URIS? Using a reasoner would make wikidata instances of type skos:Concept when they are not originally defined us such. In addition, the mappings with Eurovoc are defined with dct:subject when in this case they are originally SKOS Concepts.

The class series is subclass also of schema:Dataset in the online version but not explained in the paper. Which is the reason to define the Series as subclass of schema:Dataset and skos:Concept? Why schema instead of a vocabulary as DataCube which is actually mentioned later in Section 4.5. Actually, "RDF Cube vocabulary" a reference to the vocabulary would help to make sure whether it is data cube.

The ontology RDF files lacks metadata like publisher, licence, authors, version... (See https://lov.linkeddata.es/Recommendations_Vocabulary_Design.pdf or https://w3id.org/widoco/bestPractices or https://arxiv.org/abs/2003.13084)

The ontology is not registered in ontology registries or repositories to increase findability.


Minor:

The ontology URI does not load in Protege.

Even though the paper is in general well written, in the first paragraph in section 3.5 it is not clear whether authors refer to data or the ontology. The mention "vocabulary" in the text and "ontology" in the title but the explanation is about the data.

The state of the art only mentions a SDGIO ontology however a comparison with the datasets used for linking, which also define the SDOs could have been added.

There is an issue with the format of last words in page 9 and in page 11 "¡Goal, Target, Indicator¿". It seems to be an issue with "<" and ">" format due to a LaTeX template change.

Section 4.2 include keywords like "licensing", "Sustainability" in the middle of the paragraph, like it was a bullet list but without formatting. It does not help to find the keywords, maybe putting them in bold would met their objective.


**Anonymity:**

Yes, I would like my review to remain anonymous.

**Rating:**

-1: Weak Reject

**Strong Points:**

1) supported by UN
2) topic relevance
3) potential sustainability (not mentioned but could be related with the fact that it is a UN development)

**Subreviewer:**

I submitted this review.

**Weak Points:**

1) Modelling decisions
2) Mappings design
3) Insufficient comparison with similar resources

---

> ### Author Rebuttal · Authors · 2021-01-29
>
> We thank the reviewer for the feedback and detailed observations.
>
> We have to object about the modeling. Goals, Targets and Indicators are specific entities. Modeling them as classes would be inappropriate. In some cases, an object of the domain could be both a class and an individual and thus a choice has to be taken (unless punning is an option), depending on the perspective on the domain and on the specific representation needs. However, this is not the case: if a resource represents a single specific entity (be it concrete or abstract) there is no rationale for modeling it as a class, what its instances would be then? In the past (talking about more than 10 years, before the advent of SKOS as a W3C recommendation in 2009), many datasets have been modelled with classes (e.g. many so called “medical ontologies”, or, as we recall, Agrovoc in its first years). Silly as it can be, it was for trivial reasons: they needed a hierarchy and having confused tools such as Protégé with what should have been “dedicated visualization applications” led their maintainers to using classes. Nowadays (after the advent of SKOS), Agrovoc is a SKOS-XL thesaurus, and many medical ontologies have been converted to SKOS as well. Some might suggest that a Goal/Target/whatever could hold as instances the set of actions that aim to satisfy its objectives. However, that’s a long (and wrong) shot, and the Goal wouldn’t be a goal anymore, rather, as said, a collection of actions aiming at satisfying it. This would be better and easily represented with a dedicated class for the collection of actions and a class axiom binding that collection class to the goal itself (which would still be properly modeled as an individual). The reason for representing that individual as a skos:Concept is that this is exactly what SKOS is for: representing hierarchies of concepts, where the hierarchy is not expressing set-oriented containment of instances. Indeed, even the broader/narrower naming of SKOS is just a convention, as SKOS clearly encourages the use of these properties merely for visualization needs in representing a hierarchy of objects. The SDGs are natively modeled as a hierarchy and SKOS is definitely the modeling language to be adopted. Furthermore, SKOS’ various other properties (such as notation, with the support for multiple, differently typed, notations, just to mention one) perfectly fit the needs of such a dataset.
>
> Similar considerations hold for the series: indeed _sdgo:Series_ is a class and it contains, as instances, all the series. Each series is connected to one or more Indicators. There’s a domain property for it: _sdgo:isSeriesOf_, yet making it a _rdfs:subPropertyOf_ skos:broader thus enabling visualization in a hierarchy, as this is usually represented even before the dataset was ported to RDF. About contrast with Datacube, as the reviewer knows, SKOS concepts are the perfect candidates, when used in combination with the Datacube vocabulary, for representing discrete dimensions of statistical hypercubes.
>
> Concerning the alignments, everything can be limited to the scope of trust and believes that exists within a certain dataset. As long as a reasoner is within the KOS, a “mention” of DBPedia entities does no harm. Indeed, these external entities would be inferred to be concepts (this is indeed something debatable about setting domain and range for skos mapping properties, which are ultimately just inherited by the skos property skos:semanticRelation, which is in turn probably the original sin committed in developing SKOS) but they would just be interlingua elements for. e.g. performing alignments, never harming unless the full dbpedia is considered within a dataset, together with the SDGs.\
> Furthermore, while inferring that something is a concept mostly does no harm (it’s an additional type and has no semantic implication for a domain ontology or mere data) using the OWL mapping properties, e.g. owl:sameAs, has much stronger implications (and it - in any case - entails dbpedia resources being concepts), as mentioned in the SKOS reference (https://www.w3.org/TR/skos-reference/#L4858 ) and in the SKOS primer (see: https://www.w3.org/TR/skos-primer/#secmapping).
>
> We will fix reference to Data Cube (thanks for spotting it and apologies for the confusion) and add metadata to the ontology vocabulary (we only did it for the dataset, with the VoID/LIME metadata file).
>
> Thanks also for suggesting registering in various registries. We have already registered SDGO to LOV and SDG KOS on the LOD Cloud. They might take some time to add them to their indices, however the process has been performed.
>
> About the issue with Protégé, we are investigating it. The ontology is indeed available on the Web and VocBench imports it seamlessly. We performed another test with TopBraid Composer and it also went well.
>
> We acknowledge all the following minor observations, which will be fixed in the paper (thanks once more for this as well).

---

### Official Review · AnonReviewer2 · 2021-01-17
**A developed resource that will be used accross a wide organization, which needs improvement on the reporting of the work**

**Rating:** 1
**Confidence:** 4

**Review:**

The article entitled "A Knowledge Organization System for the United Nations Sustainable Development Goals" describes a formal knowledge organization system (KOS) which represents the United Nations Sustainable Development Goals (SDGs). The aim of the authors is to develop an ontology that models the core elements of the Global SDG indicator framework, which includes 17 Goals,169 Targets and 231 unique indicators, as well as more than 400 related statistical data series maintained by the global statistical community to monitor progress towards the SDGs, and of a dataset containing these elements.


- Have the authors evaluated the proposed interface at "http://linkedsdg.apps.officialstatistics.org/" ?I do agree that visualization is a good idea, however the model used is not necessarily suitable.



**Anonymity:**

Yes, I would like my review to remain anonymous.

**Strong Points:**

Overall the paper is well motivated and technical implementation details given. It benefits from authors expertise on the field of knowledge resource development and management. The developed resource will have a real impact throughout the United Nation organization.

Providing an implemented resource that will be used across a wide organization.

**Subreviewer:**

I submitted this review.

**Weak Points:**

Some concerns that need to be addressed in order to improve the quality of  the paper.
The introduction, while clearly stating the addressed issue, should include a survey of related work using the same approach/techniques and provide an overview of the current research in the field. In particular, claims regarding FAIR data principles (https://www.nature.com/articles/sdata201618).

Throughout the paper, there are statements with unclear fragmentary formulations, mainly because of too many partial explanations, for example: How the mapping process to external ontologies/knowledge resources like wikidata, SDGIO is carried out. This makes the paper tedious to read, quite difficult to understand.

Many Bibliographical references are out of date and do not fully describe the work in this area. Except one from 2017. There is lack of comparison with more recent/new methods.


Here follows a non-exhaustive list of typos/comments and remarks:
- Caution Fig 3 on page 13 exceeds the margins
- In section 4.1 (Usability): caution to (e.g. ¡notation¿.¡description¿)
- In section 4.3 (VoID/LIME Description): caution to (¡Goal, Target, Indicator¿ only, etc..)
- Missing space in the title of section 4.5 LinkedSDG

---

> ### Author Rebuttal · Authors · 2021-01-29
>
> We thank the reviewer for the provided feedback.  We can integrate the improvements suggested by the review, from mere typos to the more elaborated ones such as more insights on the mapping process and related works, FAIR in particular (e.g. how this KOS satisfies FAIR requirements). We believe these changes would not consist in a large rewriting of the work and could thus be suitable (if the work is accepted) for a revised version for this conference.

---

### Official Review · AnonReviewer5 · 2021-01-18
**Highly relevant application but many flaws w.r.t. the criteria for a resource**

**Rating:** 1
**Confidence:** 5

**Review:**

Update after authors' response:
* I do think that it is useful when a resource paper explains how the resource was created, not just how it should be reused. Such stories may inspire creators of further resources.
* I also stand by my point that works that are related w.r.t. applying SKOS should be mentioned. You know about them, as the discussions with the other reviewers show.
I agree that all these points can be addressed with little effort and thus change my score to a weak accept.

---

This paper presents how SKOS and its support by the VocBench system have been adapted to support the UN with information retrieval and related services in keeping track of the implementation of their Sustainable Development Goals (SDGs) with a dedicated KOS. The paper introduces the political context, the history towards the KOS, the design of the KOS itself, and an evaluation of its impact w.r.t. some of the review criteria for resources.

This development has official UN support and the SKOS-based modelling and the demo application are technically well done, but
there are other technical shortcomings. The paper, on the other hand, devotes a lot of space to the explanation
of not so relevant details of political processes. The resource was created by a solid application of state-of-the-art
techniques and thus has a good quality, but it not particularly innovative.

These are the most serious issues; please see the annotated
PDF at https://www.dropbox.com/s/3zlnjdwkqp9y3mt/a_knowledge_organization_system_for_the_united_nations_sustainable_development_goals.pdf?dl=0 for further minor details:
* Related work is entirely missing. Surely similar SKOS concept schemes have been created, and used to drive
metadata extraction applications.
* I fail to see novelty both from the SKOS and semantic web technology perspective. Linguistics is given special
attention with VoID/LIME, but it's basically all about applying existing techniques to different real-world
concepts, i.e., the SDGs.
* The alignment of the SDG ontology with SKOS is well done, but it is not clear why both variants of Dublin Core are needed
* The LinkedSDG demo at https://linkedsdg.apps.officialstatistics.org/ does not work: Browsers say "this site can’t provide a secure connection – linkedsdg.apps.officialstatistics.org uses an unsupported protocol. ERR_SSL_VERSION_OR_CIPHER_MISMATCH". I did point this out in a review of an earlier version of this paper. OK, then I tried it HTTP instead of HTTPS and found impressive how much useful information both for humans and for machines the application provides. Nevertheless, not all of this is explained well (e.g., the colourful wheel in Fig. 3), and the example in the surrounding text is a different one (water vs. women).
* Regarding reproducibility, it is not explained how the SKOS scheme was created. In what format did you take
the SDGs in order to create a SKOS scheme from them – or was it all done manually? Also, how were the links in
Section 3.5 generated? What tool was used to generate human-readable documentation from the SKOS
implementation? How were the VoID statistics generated? The paper merely says "The project
related to the development of the SDG data and the data itself (available as
download dumps) are available under a public GitHub repository" (https://github.com/UNStats/LOD4Stats/wiki), which is indeed the case, but it should be more explicit in the paper itself.
* Sustainability is good thanks to the support by the UN; however, from a more technical perspective there are
gaps. E.g., what is the roadmap for adding further languages?
* The compatibility of the license with standard licenses (e.g., CC-BY or BSD with advertising clause) should be discussed
* The quality of writing is not really bad, but there are multiple minor issues (see the PDF), and more space
should be devoted to what is actually of technical interest. The political background is relevant, but not at the
level of detail covered by the paper. Also, there is a lot of redundancy between Tab. 1 and Fig. 1. All this space
could be used for more interesting information.

**Anonymity:**

No, I would like my review to be deanonymized.

**Strong Points:**

* a really relevant solution, which helps the UN to solve a real problem
* solid application of state-of-the-art technology
* rich visual interface

**Subreviewer:**

I submitted this review.

**Weak Points:**

* no self-contained, explicit description of how the resource was produced
* little novelty against the state of the art of applying SKOS and related standards
* no discussion of related work in the sense of, e.g., SKOS applied to achieve similar things in different domains
* URL to demo is broken (HTTPS configuration issue)

---

> ### Author Rebuttal · Authors · 2021-01-29
>
> Thanks a lot for the review and for the detailed annotations in the paper that you linked on dropbox. They will really help in improving the paper.
>
> We did not put an explicit session on “related works”, even though these are mentioned in section 2 and on section 4.6. We did not think about going too further from the topics of SDG.
>
> While it is true that there’s no particular novelty, we applied for the resource track exactly because this this SDG dataset (and ontology) can represent an important resource and a reference to it; that also motivates, in our humble opinion, more space devoted (still not much, just 1,5 pages of section 2) to the political and historical background.
>
> The KOS has been generated by ingesting and converting to RDF the dump of the SDGs database to a spreadsheet file. The Sheet2RDF tool of VocBench has been used for the conversion, URI generation according to the policies described in section 3.4 etc.. We honestly thought that describing the data and its policies was more relevant than the source and the way the information has been extracted and converted. However, we can mention this process by making space for it, e.g. by removing table 2 which, as the reviewer suggests, is redundant with figure 1. We can also provide more insights about the production of the mappings, which has been mostly manual.
>
> Besides the above responses, we agree on all the technical notes both in the review here and in the linked annotated file and we have applied (or will soon do) all suggested changes, which do not require a heavy reworking of the paper nor of the dataset.

---

### Decision · Program_Chairs · 2021-02-23

**Decision:**

Accept

**Comment:**

The paper presents a KOS for an ontology for the core elements of the United Nations Sustainable Development Goals and mappings, e.g., to the United Nations Bibliographic Information System. The reviewers agree that the paper provides a unique and valuable perspective and an interesting application area for Semantic Web technologies to real and relevant problems. In terms of more critical comments, reviews and authors discussed modeling decisions, the fact that the paper is more geared towards organizational issues and not an ontology paper in the classical sense, the fact that the paper suffers from various (minor) formatting issues, and so on. Reviewers also questioned the work's novelty. I believe that this is a borderline case and that the reviewers strongly weighted the origin and domain of the paper/ontology at the expense of other aspects. Nonetheless, I suggest following their advice and accept the paper.